# WPFD: Active User-Side Detection of Evil Twins

**Fu-Hau Hsu** [1] , **Min-Hao Wu** [2,*] , **Yan-Ling Hwang** [3], **Chia-Hao Lee** [1], **Chuan-Sheng Wang** [4] and **Ting-Cheng Chang** [2]

1 Department of Computer Science & Information Engineering, National Central University, Taoyuan City 320317, Taiwan

2 Information Engineering College, Guangzhou Panyu Polytechnic, Panyu District, Guangzhou 511400, China

3 Department of Applied Foreign Languages, Chung Shan Medical University, Taichung City 40402, Taiwan

4 Chunghwa Telecom Co., Ltd. Telecommunication Laboratories, Jhongli City 32001, Taiwan

* Correspondence: mhwu@csie.ncu.edu.tw

**Abstract:** The bothersome evil twin problem has an active user-side remedy in the form of the Wireless Packet Forwarding Detector (WPFD). The evil twin issue can lead to further security problems, including man-in-the-middle (MITM) attacks. Open public Wi-Fi connections have provided potential answers to this issue, although they often need more data that people either cannot get or are too pricey for regular users. The solution that we created does not require these standards. It allows users' notebooks to be used to check for evil twins. We have succeeded in developing a user-side detection system that can successfully identify the presence of an evil twin. The packet forwarding behavior generated by the evil twin and the TCP/IP (Transmission Control Protocol/Internet Protocol) protocol are both used by the WPFD. It can identify evil twins without a hitch when we utilize accessible Wi-Fi settings in public spaces or IoT smart homes with unencrypted WLANs (Wireless Local Area Network). However, neither additional data nor a wireless network administrator's assistance is needed. We compare our work to various publications on popular Rogue Access Points (APs) or IoT (Internet of Things) smart homes. The WPFD does not require any extra setup to install on the host of any end user. According to experimental findings, the WPFD true positive and true negative rates are 100% even when Received Signal Strength Index (RSSI) is 45%.

**Keywords:** Wi-Fi; evil twin; rogue access point; wireless security; WLAN; IoT

## 1. Introduction

Access points (APs) are major devices for mobile device users to connect to the Internet. The popularity of mobile devices and the rapid development of related applications have made the ability to connect to the Internet anytime and anywhere a critical requirement of modern communication. Given this demand, the number of Wi-Fi APs is also multiplying. Many people connect to the Internet through hotspot APs in different places and locations in everyday life. An attacker can steal sensitive information from normal users through an evil twin, or launch man-in-the-middle (MITM) or phishing attacks. The popularity of mobile devices and related applications dramatically increases the demand for wireless communication. The large user pools of the hotspots make their APs attractive targets of evil twin owners. However, current evil twin solutions encounter a number of problems. Some of the solutions need the help of wireless network administrators, others require using special devices continuously, and some are easily bypassed. Hence, we plan to utilize the fact that an evil twin forwards packets to its good twin to develop two client-side solutions. We proposed an active user-side solution that could detect evil twins using only one WNIC (Wireless Network Interface Controller) to improve convenience and efficiency.

According to a study by Norton, 68% of people who use public Wi-Fi networks are victims of cybercrime, mainly the theft of sensitive data, including passwords, bank account information, credit card numbers, chat logs, and emails [1]. Public networks are susceptible

to several types of attacks, including evil twins, since packets are sent over the air. The adversary known as the "evil twin" poses as a lawful access point (LAP), which is possible by forging the MAC (Media Access Control Address) address and SSID (Received Signal Strength Index)of the LAP (Legitimate Access Point) (BSSID, Basic Service Set Identifier). An adversary's access point may unintentionally be linked to by a user who mistakenly thinks the access point belongs to the whole network. When establishing a connection, attackers can set up man-in-the-middle, service disruption, and access point denial-of-service attacks [2]. It is simple for an attacker to set up an evil twin attack in public places when the network utilizes open Wi-Fi. The first stage involves the attacker installing software on a PC (Personal Computer) or a Raspberry Pi and configuring the evil twin AP in a Wi-Fi network. By setting up a device with the same Wi-Fi name (SSID) and MAC address (BSSID) as the genuine AP, the fake AP may trick clients into connecting to it—that is, to the attacker—by pretending to be the real AP. The next phase involves dishonest opponents using directional antennas to lengthen their signal or to boost the Received Signal Strength Index (RSSI) by positioning the malicious twin APs closer to the consumer than genuine APs [2,3]. As a result, clients may be reluctant to connect to the evil twin AP when they wish to utilize an approved LAP to access the Internet. An enemy may also use the evil twin AP to monitor all client network activity [3].

On the part of the network provider, network managers are in charge of helping wireless customers spot evil twin assaults. ETA (Estimated Time of Arrival) detection is costly to maintain for a Wi-Fi network. It could be necessary to install wireless sensors in routers that collect data for comparison with certified lists of distinguishing traits that are now available on the market [3]. When Pragati Shrivastava et al. [2] examined the evil twin attack on an actual testbed with varied Wi-Fi settings, they discovered that it might result in a wide range of assaults, including MITM, service interactions, and other types of attacks. Pragati Shrivastava et al. [2] adapted, tested, and implemented ETs in Wi-Fi networks supporting Software Defined Networking by using the information already present in the Wi-Fi networks (SDN). The clock skew approach is the next management-party evil twin assault detection method. In Wi-Fi networks, Jana et al. [4] used clock skew fingerprinting, in which the timestamp provided in the beacon frame exhibits a unique physical fingerprint. By comparing the target AP's clock deviation with the clock deviation of other accessible APs, the evil twin APs might be distinguished from genuine APs. A passive client detection technique was presented by Qian Lu et al. [5–7] that allows clients to detect evil twin APs without the help of the wireless network administrator. The technique analyzes the forwarding frame and the forwarding behavior of the evil twins. It could demonstrate the method of comparing the wireless data frames sent to the client by the target AP. This proposed passive scheme enables the client to identify and locate the ETA and decide whether to connect to ETAP (Endoscopic Transanal Proctectomy) or LAP (Laparoscopic Proctectomy). The operating system has two kinds of networks [5]: a preferred network list, which is a list of the client's prior connections, and the list of networks that are accessible, along with the list of first-connected clients. The suggested solution, which solves the issue mentioned earlier by using the operating system's potential, is implemented on the side of the access point. At the same time, it is executed on the operating system. The second segment could be able to uncover attacks by the evil twin [5,8].

Attackers can steal user information through an evil twin. However, existing solutions to this problem require system management. The users' collaboration either involves long-term usage of specific equipment or is readily bypassed. We developed a solution that does not have the above problems. Users can use their laptops to detect evil twins at any time. We have successfully created a user-side detection mechanism that can effectively detect the existence of an evil twin. The detection mechanism does not require additional features of detection devices. That is, we can reduce the cost of detection at the user-end and increase detection efficiency. We consider it comprehensive protection.

In eavesdropping, wireless communication is airborne. It is simple for attackers to collect and store data moving through the network. Even though the communication is

encrypted, it is still critical to be aware that attackers may examine the received portion of the message and gather crucial information. For their attack requirements, attackers may disseminate specific messages over the network. An attacker can also discover the encrypted key and use it to decrypt other packets through the information in the recorded packets or the clear text. Attackers utilize their devices to passively eavesdrop on wireless network packets and analyze the network conversation traffic, content, and behavior. Communication content or traffic analysis may discover information about the target network, such as the server IP and communication mode. Since the attacker observes the packets moving on the web in front of the computer, no trace is left. It is possible to intercept and store the content of communications by using the monitoring mode with software like Kismet. Some listening software may simulate an analog terminal to show the communication's content, allowing the listener to view the screen precisely as the user would.

A rogue AP may be an evil twin. Typically, a traditional Rogue AP is installed by an insider and connects directly to the internal wired network. In contrast, an evil twin is installed by an outsider and links to the Internet through an available AP (called good twin). Evil twins are depicted in Figure 1. Additionally, attackers can launch de-authentication attacks to disconnect normal Wi-Fi users from good twin APs and connect them to evil twin APs.

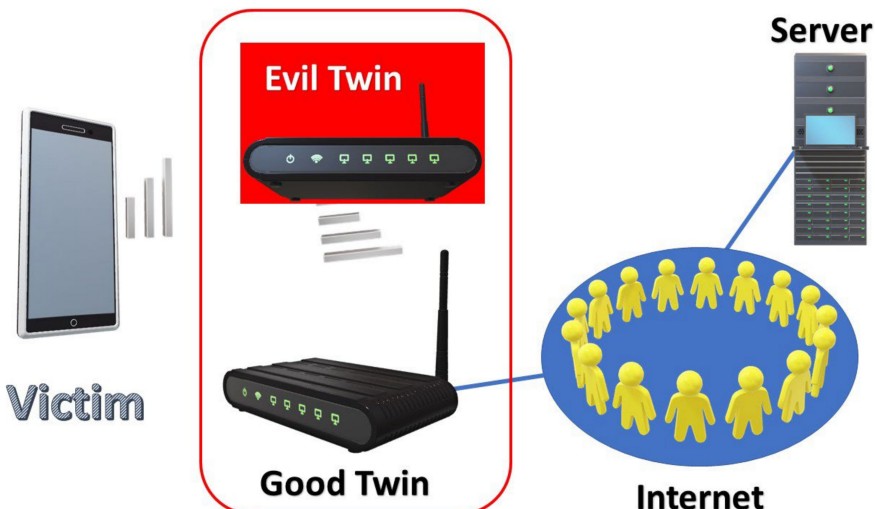

**Figure 1.** A scenario of an evil twin setup.

An evil twin can track the data and steal important information like user passwords and personal credit card details. If a client uses HTTPS to connect to a website, he is still at risk of MITM attacks. Once a client connects to an evil twin, it is far easier for attackers to steal information using techniques such as an SSL (Secure Sockets Layer) bump. A VPN may solve this problem; however, it is too expensive to use VPNs for every network service. Moreover, according to [9], a malicious attack by an evil twin is easy to launch. Most of the solutions to evil twin problems were designed with a system administrator's permission. The main disadvantage of this approach is that the defenders may need (1) one specific network device for detection, such as one wireless sensor or router, (2) some particular data, such as legal APs/IPs or training data via machine learning, and (3) network traffic trace information. The above requirements are usually not available to general Wi-Fi users.

Due to the problems mentioned above, this paper proposes an active client-side solution named Wireless Packet Forwarding Detector (WPFD) to solve the issues regarding some free public hotspots such as unencrypted WLANs. WPFD transforms the traditional detection approaches from the primary device-based detection methods to TCP/IP protocol-based monitoring methods, with packet retransmission rules and packet-forwarding conventional behavior. Thus, the only information required is the TCP/IP header of the

wireless packet. WPFD sends SYN (Synchronize sequence numbers) packets to popular websites and observes the retransmission behavior of corresponding SYN/ACK (Synchronize sequence numbers/Acknowledgment field significant) packets. Our consistent results show that WPFD has an accurate and fast detection capability for the presence of an evil twin after observing only a few packets. The features of WPFD make it an appropriate tool for Wi-Fi users. They are as follows:

1.  The WPFD does not require specific information, such as legal APs/IPs, training data via machine learning, or any other support from a network administrator. The WPFD is a user-side solution that may secure any Wi-Fi user at any location.
2.  The WPFD sends SYN packets to significant websites to make its detection a more effective solution. An evil twin finds it hard to detect that it is being monitored by a WPFD, and is even less likely to adopt any anti-surveillance action in time.
3.  It is based on evil twins and TCP/IP protocol. An evil twin will barely evade discovery by a WPFD. Hence, even if an evil twin can raise the packet delay time or manipulate the content of IP packets, it still needs to forward probing IP packets sent by a WPFD. A probe IP packet is just a regular SYN packet.
4.  The WPFD detection accuracy is unaffected by topology, network type, traffic, or any prefetch mechanism because it does not rely on time measurement.
5.  The WPFD is the first solution that combines the active probe packets and the passive monitor mode of a WNIC (Wireless Network Interface Controller) using only one WNIC and no custom server. As a result, WPFD is well suited to solving the evil twin problem in devices with only one WNIC.

The related work on rogue APs and evil twins is discussed in Section 2. The principles and the algorithm of the WPFD are described in Section 3. The discussion on various experimental results, and the evaluation of the effectiveness and efficiency are given in in Section 4, along with the debate on the security issues related to WPFDs. Finally, the conclusions of our study on WPFDs are in Section 5.

## 2. Related Work

Because of the severe threat posed by rogue access points, industrial and academic researchers have proposed many solutions. Radiofrequency sniffing methodology uses a variety of devices, such as sensor APs and sensors [10,11]. These are based on comparing the fingerprint to the authorization list to filter out rogue APs. Nevertheless, the above solutions do not offer complete protection for WLANs or their users without continuous monitoring, since attackers can quickly build and remove malicious twins anytime. Additionally, these solutions may misidentify legitimate neighbor APs as rogue APs. Some hybrid solutions have been proposed to overcome the above problems [12–14]. For instance, Bahl et al. [12] turns into a wireless sniffer to reduce deployment costs and improve efficiency. Lu et al. [13] and Yin [14] does not sniff wireless traffic. Therefore, if internal sensors observe the duplicate packets in wired traffic after they send packages to the Internet through a suspicious AP, the system can determine if the suspicious AP is an evil twin or not.

Furthermore, Beyah et al. [15] were the first authors to detect rogue APs passively using the previously mentioned time metrics. Using local RTT (Round Trip Time), Mano et al. [16] distinguished wired traffic from wireless traffic. On the other hand, client-centric evil twin AP detection is a new solution class for evil twin AP detection [9,17–20]. Such solutions are ideal for quick connections wherever Wi-Fi access is required for AP travelers. They do not require any permitted IP list of APs, WLAN operator support, or network IP tracking via gateways.

Nonetheless, this solution also faces some common challenges that must be overcome before it can be widely adopted. First, active detection requires the user to connect to the Internet before detection occurs. But many applications now automatically log into service after connecting to the Internet, which means that an attacker may have obtained the user's essential information before the exam has started. Second, quest packs often come in particular forms that are easy to spot by the evil twin.

Nicholson et al. [21] proposed Virgil to automatically discover and select access points. Virgil associates with each AP discovered during the scan and chooses the best one based on bandwidth estimates and round-trip times to a set of reference servers.

Yan et al. [9] also proposed ETSniffer, a client-side Evil Twin AP detection system that uses Inter-Packet Arrival Time (IAT). Because it does not require any assistance from the WLAN operator or a list of authorized Aps or hosts, ETSniffer is ideal for travelers. However, ETSniffer must send a specific IP packet per the immediate-ACK policy to obtain accurate IAT data. As a result, the evil twin can detect the packet being inspected based on its structure, and avoid detection by employing various techniques such as prefetching web page content.

Chatzoglou et al. [22] proposed that there are reasonable criteria to determine. Based on theoretical and empirical observations, the MAC layer characteristics are the most informative and deployment-independent features for training wireless IDS (Intrusion-Detection System), and have the lowest number of features that may produce at least fair detection rates. The study shows that accuracy criteria are extensively used in earlier work to assess IDS models. It is also a result of using a balanced dataset, as previously discussed in this section. We think this is primarily because of the features used in the tests that could utilize only a small number of characteristics in practice. For each model, the score, accuracy, and overall execution time (in hours, minutes, and seconds) are considered.

Shrivastava et al. [2] discovered a Wi-Fi client connected to the LAP and the evil twin in terms of wireless coverage when the evil twin impersonates the LAP by spoofing its BSSID on the same channel. As a result, APSB (AP Service Blocking)attacks are made feasible by interfering with the client's EAPOL (Extensible authentication protocol over LAN) 4-way handshake and the WPA2-protected LAP, taking advantage of the evil twin's ability to operate on the same Wi-Fi channel as the LAP. EvilScout is installed on top of the SDN controller and takes advantage of OpenFlow features like PacketIn and writes FlowMod messages. The SDN Wi-Fi architecture enables developers to include new capabilities but also creates unknown security risks. For instance, because the controller is the brains of SDN Wi-Fi, there should be no design or implementation errors, and Duplicate Association is used to spot nefarious twin assaults. Since EvilScout only needs to verify one packet from every victim client, it has little controller overhead and is capable of quick detection. The research offers a thorough knowledge of APSB assaults and potential defense mechanisms.

The most novel and standard part of the application of Wi-Fi communication in recent years is the IoT, and the typical example of an IoT will be the smart home. For example, we can use Wi-Fi to build a smart home. Then we can use wireless communication to establish simple household operations of kitchen items such as refrigerators or electrical appliances (the most common ones are televisions, speakers, etc.). In this case, if the Wi-Fi is considered only for self-use at home without unique encryption settings, under this application, such an innovative home environment may also be exposed to the risk of malicious evil twins. That is to say, in a public domain where free Wi-Fi is provided and in the private area of the home with poor security policies, the evil twin will also obtain personal information in practice if an attacker deliberately locks someone. Also, the risk is even greater if IoT devices are surveillance systems and cameras closely related to privacy. Sruthy et al., who studied Wi-Fi-enabled home security surveillance systems using Raspberry Pi and IoT module, mention the connected novel system. However, such a system is also threatened by the evil twin. Even the smart meter, as in Win Hlaing et al. [23], will be exposed to this threat. The attacker can get their information about the power wastage and cost of consumption of the house. In fact, for this kind of research on Rouge AP to IoT, Agyemang et al. [24] presented a real-time and lightweight algorithm based on an information-theoretic approach for detecting rogue APs in embedded IoT devices.

The most directly related to our work was the solution proposed by Monica and Ribeiro [25], called Wi-Fihop, which detects evil twins based on their forward behavior. Wi-Fihop sends active probe packets and switches the WNIC to the monitor mode. It analyzes the response packets to determine if there is an evil twin. This approach is

similar to our solution, and Wi-Fihop also needs one WNIC on the client side. However, unlike our solution, which is a purely client-side tool, Wi-Fihop requires a customized server to respond to specific probe packets. Furthermore, the particular probe packets are easily detected by an experienced attacker who can ignore the probe packets to bypass the detection of Wi-Fi. The WPFD does not have this problem

On the other hand, in the previous research on general rogue APs there is a summary or a discussion like Sweta et al. [26], and for a round-trip time measurement with a machine learning algorithm, in Songrit et al. [27]. Furthermore, as mentioned above, there are also algorithms like Ganesh et al., which is a standard algorithm of the LAN network environment [28], and Neha et al. [29], which is a hash algorithm focused on WLAN security. Finally, for the Wireless LAN network, there is a detection solution designed by Sandeep et al. [30] with a sensor node and a heartbeat method to prevent Rogue APs. Table 1 compares the accuracy performance of several evil twin attack detection systems.

**Table 1.** Comparisons of different evil twin attack detection techniques.

| Technique | Evil Twin Attack Detection Systems | |
|---|---|---|
| Duplicate RSSI | Client Side | Passive |
| Clock Skew | Client Side | Passive |
| Context-leashing | Client Side | Active |
| Traffic Monitoring | Network | Passive |
| DNS Server | Network | Active |

## 3. Design Principle and Detection Algorithm

This section describes the wireless network interface card (WNIC) monitoring method, the fundamental phenomenon of an evil twin attack, and the WPFD design principle and detection algorithm.

### 3.1. Monitor Mode

A WNIC in monitor mode will monitor all nearby wireless traffic. Unlike the promiscuous mode, monitor mode allows the WNIC to capture all wireless packets while not connected to an AP. Most WNICs and modern operating systems support monitor mode. However, everyone has their method of transitioning to this model. Microsoft Network Monitor [31] enables users to operate the WNIC in Windows efficiently. In the Unix operating system family, activating monitor mode is more straightforward. There are some networking-related built-in commands for this purpose.

In general, because the evil twin needs to connect to the Internet through the good twin, the evil twin must have two wireless adapters. Users may be tempted to believe that the adaptor is an official AP. Because this adapter has an SSID, the WNIC will recognize it as an AP. The other is used to connect the good twin. As a result, it lacks an SSID and functions similarly to a handheld device or a notebook. Even if an evil twin and the corresponding good twin have the same SSID, their MAC addresses or BSSIDs are different (Basic Service Set IDs). An attacker can impersonate a good twin's BSSID. The assumption, however, is that the two devices share the same BSSID. In that case, they will process all packets sent to that BSSID simultaneously, resulting in the disconnection of a TCP/IP connection between a server and a device with the BSSID. As a result, if the evil twin has the same BSSID as the good twin, it can no longer complete its attack. Figure 2 depicts the WNIC's perspective on an evil twin.

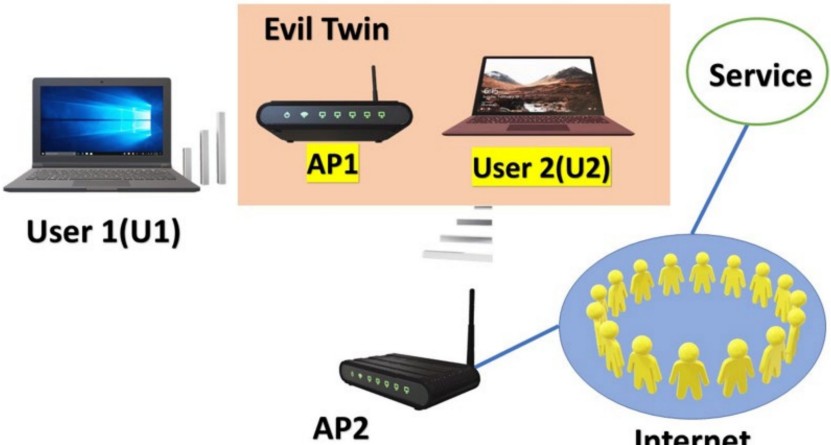

**Figure 2.** The perception of an Evil Twin by a WNIC (Wireless Network Interface Controller).

### 3.2. Packet Retransmission and Packet Forwarding

In this paper, we utilize the monitor mode of a WNIC to detect the packet forwarding phenomenon and determine whether an AP is an evil twin or not. If an evil twin forwards IP packets directly, an original IP packet and the related forwarded packet are supposed to be almost the same, which is an ideal feature for evil twin detection. However, a WNIC can monitor only one channel at a time. If an evil twin uses a medium different from its good twin's track, and a WNIC cannot switch between these two channels at the right time, the WNIC cannot detect both the original packets and the forwarded packets. Furthermore, even if a solution implements channel switching to collect packages on both channels, the overhead of the channel switching would cause packet loss, which may affect the detection accuracy of the solution.

This paper utilizes the retransmission mechanism of the TCP protocol to solve the above problems and detect the packet forwarding behavior. The packet retransmission mechanism is a fundamental TCP/IP protocol rule, providing reliable data transmission. It is difficult, if not impossible, for an attacker to intervene in the retransmission mechanism, especially in public hotspots where packet retransmission could happen frequently. The WPFD is based on this unchangeable property, making it hard to evade its detection mechanism. Even if rogue APs could change the content of forwarded packets, they cannot change packages as in a three-way handshake, because changing these packets will destroy the integrity of the TCP connections.

### 3.3. The Principle and Wireless Packet Forwarding Detector (WPFD) Algorithm

The WPFD can be installed on any host without any special requirements. WPFD detects evil twins based on the general rule that a good twin will not forward wireless packets. The packet transmission process for a three-way handshake to establish a TCP connection through the evil twin is shown in Figure 3. In the fifth step, if User 1 does not send the ACK packet, the server would retransmit the SYN/ACK packet when the RTO (Retransmission Timeout) is reached. Henceforth, we refer to this retransmission of an SYN/ACK packet associated with a WPFD TCP/IP detection connection as a probe hit. Every time a packet is retransmitted, the RTO for this packet doubles. Operating systems often resend an IP packet at least five times [32], and most web servers define the initial RTO from 2 to 3 s.

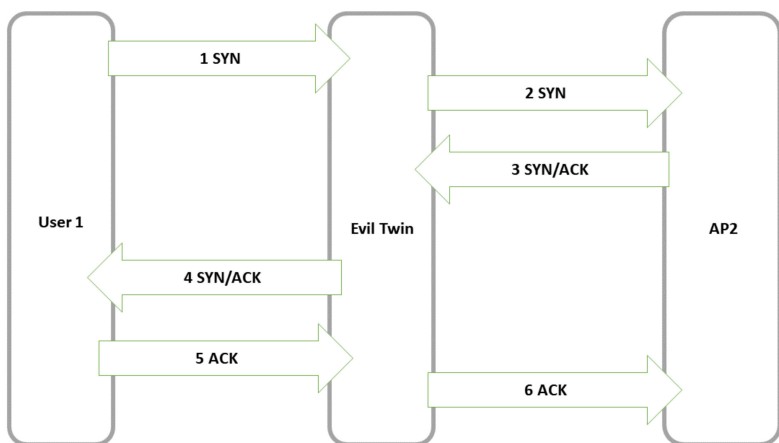

**Figure 3.** The packet sequence for the three-way handshake of User 1 was used to establish one TCP (Transmission Control Protocol) connection through an evil twin.

As described in the previous section, this paper utilizes the retransmission mechanism and the three-way handshake mechanism of the TCP/IP protocol to detect packet forwarding behavior in Wi-Fi networks. We propose the following algorithm, called Wireless Packet Forwarding Detector (WPFD) algorithm, to detect evil twins.

Step 1: Choose a remote host $IP_{server}$ with an IP address from a list of websites. The top 10,000 websites on Alexa are used to create the list.

Step 2: WPFD checks the environment to see if any APs with the same SSID are present. If not, there are no evil twins. Hence the detecting process is over. If they are present, WPFD will choose all APs with the same SSID, sometimes referred to as suspicious APs. WPFD discovers the Wi-Fi channel that each suspicious AP is using. Channel groups are used to classify suspect APs that are connected to the same Wi-Fi channel. WPFD gathers each channel group before moving on to step 3.

Step 3: Let $S$ stand for a collection of the channel groups that WPFD gathered in step 2. To determine if a suspicious AP in $S$ is an evil twin, WPFD employs Algorithm 1. The suspect AP, $AP_{input}$ is chosen as Algorithm 1's input. Algorithm 1's current inputs are $IP_{server}$, $AP_{input}$, and $S_{input}$. The return result indicates whether or not $AP_{input}$ has an evil twin.

Every channel group employs a channel, and its *Repeats* will be assessed. If $AP_{input}$ has an evil twin, it will connect to the Internet using a legitimate AP. Let $ProbeHit_i$ indicate the number of times $IP_{server}$ retransmitted when algorithm 1 is applied to channel group $G_i$ using channel $CH_i$. $Total_i$ is a counter used to determine the number of examinations performed when $G_i$ is utilized in algorithm 1, and channel $CH_i$ will be used by suspect APs. The greater $ProbeHit_i$ is, the more retransmissions occur in channel $CH_i$ which suspect APs in $G_i$ use. The ratio of $robeHit_i$ to $Total_i$ might be used to determine if $AP_{input}$ forwards $SYN$ packets sent by WPFD, since SYN/ACK packet retransmissions from a certain $IP_{server}$ do not occur often. *Repeats* and *Threshold* should be set to 5 and 0.1, respectively. Section 4 explains the rationale.

---

**Algorithm 1: Wireless Packet Forwarding Detector.**

---

Input:

 $IP_{server}$: a remote server for WPFD to connect to.

 $AP_{input}$: a suspect AP under evil twin examination.

 $S_{input}$: a set of channel groups consisting of suspect APs with the same SSID.

Output:

 *True*: if APinput is an evil twin

 *False*: if it is not;

1: Do:
2: Choice a $G_i$ from $S_{input}$;
3: Integer $ProbeHit_i$, $Total_i = 0$, $Repeats$;
4: Long $Threshold$;
5:  Do:
6:   Connect to $AP_{input}$, and send a SYN packet to $IP_{server}$;
7:   Switch the WNIC to monitor mode, and monitor channel $CH_i$ for one minute;
8:   Collect the SYN/ACK packets whose source IP addresses are equal to
    $IP_{server}$, and switch the WNIC to normal mode;
9:   if (Two or more collected packets are identical) :
10:    $ProbeHit_i$++;
11:   end if
12:   $Total_i$++;
13:  While: ($Total_i <= Repeats$)
14:  if ($ProbeHit_i/Total_i > Threshold$) :
15:   return *TRUE*;
16:  end if
17:  $S_{input} = S_{input} - G_i$;
18: While: ($S_{input} \mathrel{!}= \varnothing$)

---

## 4. Evaluation

This section will review the WPFD mechanism's effectiveness, accuracy, efficiency, and limitations. The WPFD can be installed on a notebook with a wireless Network Interface Card (NIC), a 2.4 GHz Intel Core 2 CPU, and 4 GB memory running Microsoft Windows 7 64-bit.

### 4.1. TCP/IP Connection Establishment

To validate WPFD, we used two hosts to simulate a regular user's client hosts and used these hosts to establish a TCP/IP connection to a remote service. We started by using a sniffer to watch hosts as they browsed Gmail, Facebook, 280 Google, Twitter, and other popular web services. The observed pattern will establish a TCP/IP connection to the websites www.google.com, accessed on 8 July 2022 and tw.yahoo.com, accessed on 8 July 2022. According to the findings, while browsing three websites, Facebook, Gmail, and Twitter, at least one TCP/IP connection was formed within 30 s, and at least five links were established within one minute. We begin our investigation in the following subsections with the measured TCP/IP connection rate.

### 4.2. Accuracy Analysis

We executed a mock program on the client's host machine. The program establishes a TCP/IP connection to the specified remote website service every minute. Two laptops are used in the experiments to impersonate two ordinary users. One notebook chooses the best twin to connect to the website www.google.com, accessed on 8 July 2022. The other laptop deliberately selects the evil twin to connect to the site "tw.yahoo.com", accessed on 8 July 2022. Figure 4 depicts the experimental environment for detecting evil twins. In this environment, we sent SYN packets to $AP_2$ and monitored the TCP/IP packets handled by

$User_3$ to check the good twin detection accuracy of WPFD. On the other hand, we sent SYN packets to $AP_1$ and monitored the TCP/IP packets directed by $User_1$ to prevent the evil twin detection accuracy of WPFD. We evaluated WPFD with different RSSI values in this environment. The WPFD made 1000 probes to each AP and logged the corresponding probe hits. Table 2 shows the results. RSSI values significantly affect the detection effectiveness of WPFD. However, the differences in probe hits between the good and evil twins are significant enough to identify the evil twin. The *ProbeHit* to *Total* ratio for the evil twin is always greater than 50%. However, the proportion of the good twin is always less than 2%. As a result, as described in Section 3.3, we set the *Threshold* to 10%. As a result, WPFD has a true positive rate of 100% and a true negative rate of 100%. The notebooks will be set up to imitate two regular users to open 1, 2, 5, or 10 TCP/IP connections per second to verify the accuracy of WPFD under varying traffic levels. Figure 5 shows that traffic 310 volumes have little influence on WPFD accuracy.

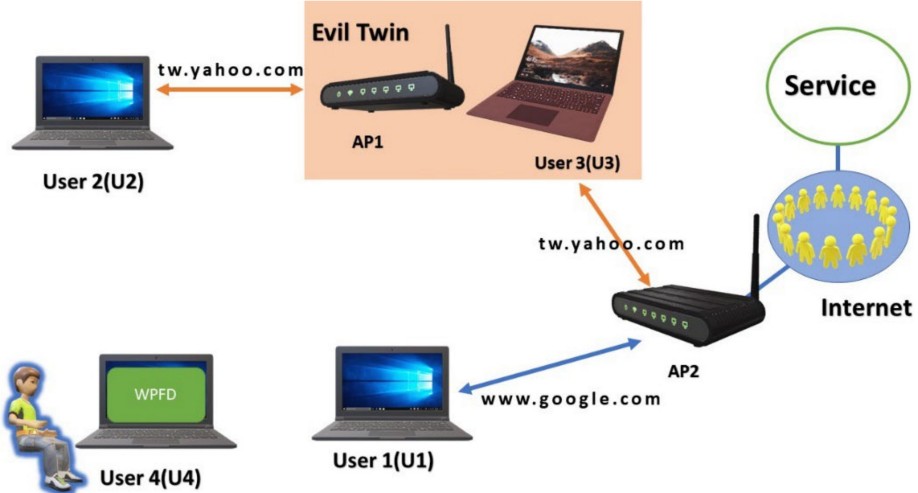

**Figure 4.** Experimental environment for evil twin and good twin detection.

**Table 2.** Effectiveness experiment results of WPFD with different RSSI values.

| RSSI to Good AP | Probe-Hits of the Good Twin | Probe-Hits of the Evil Twin |
|---|---|---|
| 95% | 19 | 981 |
| 80% | 8 | 902 |
| 70% | 1 | 985 |
| 60% | 0 | 946 |
| 45% | 0 | 507 |

### 4.3. Time Efficiency

In this section, we evaluate the system algorithm's time efficiency. We tested our system with various detection durations. Table 3 shows the outcome. The total number of packets captured is proportional to the number of connection flows detected during the detection period. Therefore, we are not concerned with the increment of complete packages but the relationship between the good twin's probe hits and the evil twin's probe hits. By observing the result, we can see that the number of probe hits of the evil twin is always more significant than the number of probe hits of the good twin. Hence, we can determine the evil twin as soon as a few probes hits occur. Each probe takes about one minute. As a result, we set the Repeats to 5, described in Section 3.3. Hence, the WPFD can determine an evil twin in 5 min.

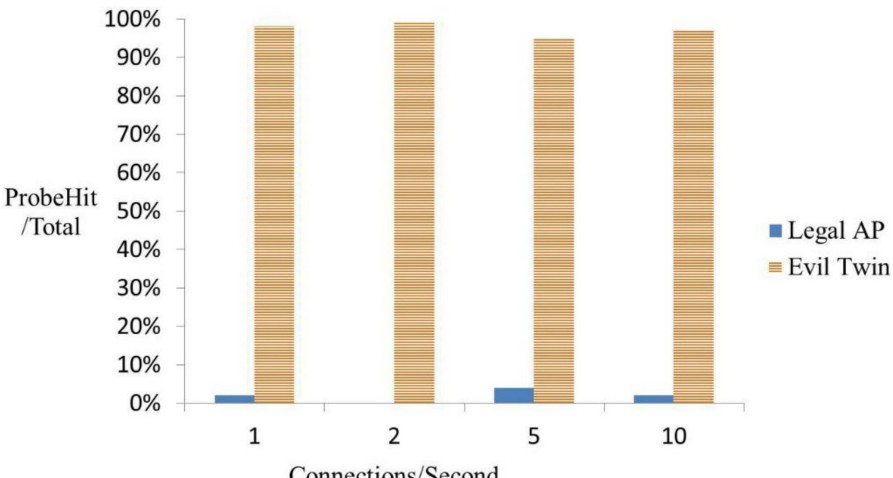

**Figure 5.** Accuracy of WPFD under various volumes of network traffic.

**Table 3.** Time efficiency experimental results under different test durations.

| Time | Probe Hits of the Good Twin | Probe Hits of the Evil Twin |
| --- | --- | --- |
| 3 min | 0 | 3 |
| 5 min | 0 | 4 |
| 7 min | 0 | 6 |
| 10 min | 1 | 9 |
| 15 min | 1 | 14 |

### 4.4. Limitations

The WPFD could detect evil twins in most situations. However, there are still some unique situations in which an evil twin could bypass the detection of WPFD. This subsection describes the scenarios. In the first scenario, an evil AP still uses a good AP to connect to the Internet. But the evil AP uses an SSID different from the good APs and attracts victims to connect to it through other methods such as karma attacks [33]. Even though this scenario is not an evil twin attack, according to the definition of evil twins, we still describe this scenario as a situation that WPFD cannot handle. In the future, we plan to solve this problem by letting WPFD observe the packet-forwarding behavior of all APs in a wireless network.

In the second scenario, an evil twin uses a cable or a 3G/4G device to connect to the Internet. A WNIC cannot detect the redirection behavior in this scenario. This problem, however, becomes a rouge AP problem, which is outside the scope of the evil twin problem.

### 4.5. Discussion

We compare WPFD with previous studies on detecting evil twins [19,20] and a traditional time-metric-based solution [34]. Hsu et al. [19] is our prototype using the evil twin redirection methodology. However, if the adversary modifies the sequence numbers and ACKs, it will be bypassed. Instead, WPFD only sends SYN packets. These probe SYN packets sent by WPFD behave the same way as standard TCP three-way handshake packets. Therefore, attackers have little chance of finding the probe packets, bypassing the WPFD's detection. Kuo et al. [34] showed that time-based methods might be affected by various environmental factors. Another difference is that a WPFD only needs one device because it uses the same WNIC to send probe packets and switch to monitor mode. When we studied LAF [20], it had some of the advantages of the WPFD. Still, the WPFD has evolved into an active detection mechanism after changing the algorithm from the passive detection mechanism of LAF. The novel dynamic mechanism is more capable of detecting

the evil twin at any time, which has excellent usability. Table 4 shows these advantages and differences.

**Table 4.** Comparisons between WPFD and previous works.

|  | WPFD | LAF [20] | Hsu et al. [19] | Kuo et al. [34] |
|---|---|---|---|---|
| Detection Time | a few minutes | a few minutes | a few seconds to a few minutes | about one minute |
| Affected by Network Traffic | little | little | little | great |
| Bypass detection | hard | hard | easy | easy |
| Usability | excellent | good | average | fair |

If an attacker could detect the probe packets sent by a WPFD, the attacker could block such packets or send RST packets to stop the retransmission of SYN/ACK packets from servers. However, a WPFD only sends SYN packets. It is not reasonable for an attacker to block every SYN packet. Moreover, a WPFD can change the source IP addresses or MAC addresses of probe SYN packets. It can also send different probe SYN packets to other servers. In this case, an attacker is brutally distinguishing the probe SYN packets sent by WPFD from the standard SYN packets sent by a regular user. Thus, the attacker cannot bypass the detection of a WPFD. Retransmission of IP packets may frequently occur in a Wi-Fi network. Therefore, users connected to an evil twin experience disconnection if the evil twin blocks retransmitted packets. In the end, users will not use the evil twin and choose a good twin to connect to the Internet.

Second, a WPFD functions well even if multiple APs have identical SSID but use different channels. However, the more channels are involved, the more detection time is needed. There could exist three or more other channels used by suspect APs in a Wi-Fi network. Thus, the detection time would increase. To address this problem, a WPFD could speed up the detection time by sending multiple probe SYN packets to numerous servers simultaneously and checking their responses. In this situation, adjusting the 5 Repeats times in the algorithm might still restrict the detection time of a WPFD to a few minutes.

## 5. Conclusions

We propose an active user-side solution, the Wireless Packet Forwarding Detector (WPFD), to the problem of a wireless user inadvertently connecting to the Internet via an evil twin. Although WPFD is an active solution for evil twins, an evil twin may be unaware of its existence. We have done many trials showing that WPFDs can accurately locate an evil twin using a few wireless packets. Compared with prior investigations, this active mechanism is more capable of recognizing the existence of an evil twin in an open space at any moment. Thus, this is the best solution to the evil twin detection issue. It is also easy to use and practical. Effective detection can be completed without a plain approach or a passive waiting method. The research findings indicate that the WPFD will be capable of solving the evil twin problem of Wi-Fi with a 100 percent true positive rate and a 100 percent true negative rate, whether in a free Wi-Fi situation or a private area of the smart home.

**Author Contributions:** Conceptualization, F.-H.H.; Formal analysis, F.-H.H.; Investigation, F.-H.H. and C.-S.W.; Methodology, C.-S.W.; Validation, Y.-L.H.; Visualization, T.-C.C.; Writing—original draft, M.-H.W. and C.-H.L.; Writing—review & editing, M.-H.W. and C.-H.L. All authors have read and agreed to the published version of the manuscript.

**Funding:** This work was also supported by the Ministry of Science and Technology, Taiwan, R.O.C. under Grant No. MOST 105-2221-E-008-074-MY3 and 111-2221-E-008-080-MY3. This study was funded by the Innovation and Entrepreneurship Training Program-Intelligent and Convenient Elderly Physical Examination System. (Research Grant no. 2011/210113263). Research on Brain Ripple Encryption Based on Emotional Speech. (Research Grant no. 1102/210339007). Lightweight authentication key negotiation with privacy protection in a medical environment. (Research Grant no. 1007/210224339).

**Institutional Review Board Statement:** Not applicable.

**Informed Consent Statement:** Not applicable.

**Data Availability Statement:** Not applicable.

**Conflicts of Interest:** The authors declare no conflict of interest.

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
