# Peer review of "WPFD: Active User-Side Detection of Evil Twins"

_applsci, doi:10.3390/app12168088_

Round 1

Reviewer 1 Report

The Authors present an algorithm to defend infrastructure-based WLAN users against the evil twin attack on the part of a bogus Access Point (AP) that might intercept the users' traffic and retrieve therefrom sensitive data before forwarding them to a legitimate AP. Contrary to some known solutions, the proposed algorithm is entirely user-side and does not require dedicated agents at the genuine AP or remote server. It amounts to active probing: a user sends control packets of the TCP connection setup and senses the wireless medium for control packets belonging to the same connection; the presence of an unwelcome forwarder (the evil twin AP) statistically increases the chance of sensing repetitions of such control packets. In the proposed algorithm, wireless channels used by APs announcing the same SSID are successively monitored for unusually numerous repetitions subject to experimentally selected monitoring duration and repetition frequency threshold. The idea seems plausible though the results and discussion are only specific to the employed laboratory testbed; the literature review is good, and the validation approach looks solid. Overall, I consider the paper publishable after two types of revision. First, the algorithm is in fact quite straightforward and so would be more comprehensible if expressed semi-formally rather than using the pseudocode laid out on p. 7, which introduces complex notation and language constructions to convey simple ideas. Second, the grammar and style are unacceptable, please have the manuscript thoroughly re-edited and checked for clarity, usage of vocabulary, and logic of thought.

Reviewer 2 Report

·       The overall quality of this paper is Good

·       The Originality of this paper is Good

·       The topic is exciting and relevant.

·       The paper organization and flow of ideas are both good.

·       The technical writing of the paper should be improved.

·       The keywords used in the paper are satisfactory.

·       Title

It should be Rephrased! And find a better one

·       Abstract

The abstract is not coherent. It would be good if authors could write a sentence describing numerical results and improvements over other methods. The abstract needs to be improved and the main contribution of the paper should be stated clearly in the abstract section.

·       Introduction

The introduction part is good. However, it can be imporived to give a general knowledge about the area with a few examples

·       Literature Works

·       More Recent related works from high-impact journals are required. The research gaps should be highlighted clearly, why this work is important, it could be better to add a table at the end of the related works section to summarize the research gaps and how this work will respond to these gaps.

·       The related work section is too small, it could not be considered as it is. More related works are required,

·       Design Principle and Detection Algorithm

o   The algorithms need to be rewritten and a good way for understating.

o   Results need more writing for an explanation, not just descriptive cases 

Round 2

Reviewer 1 Report

I find substantial work done in response to my former comments, which has improved the quality of the manuscript. In my opinion, the English is still not quite acceptable, which might be considered of little importance were it not for its impact upon the readability of the main idea. At the same time, the problem being considered is interesting and timely, as such is worth publiaction. Therefore, I would recommend minor revision, with special care given to the clarity of Algorithm 1. Also, for the uninitiated I would stress more how an evil twin attack differs from simple eavesdropping.
